# A Case of a Malignant Lymphoma Patient Persistently Infected with SARS-CoV-2 for More than 6 Months

**DOI:** 10.3390/medicina59010108

**Published:** 2023-01-04

**Authors:** Yoji Nagasaki, Masanori Kadowaki, Asako Nakamura, Yoshiki Etoh, Masatoshi Shimo, Sayoko Ishihara, Yoko Arimizu, Rena Iwamoto, Seiji Kamamuta, Hiromi Iwasaki

**Affiliations:** 1Department of Infectious Disease, Clinical Research Institute, National Hospital Organization Kyushu Medical Center, Fukuoka 8108563, Japan; 2Department of Hematology, Clinical Research Institute, National Hospital Organization Kyushu Medical Center, Fukuoka 8108563, Japan; 3Fukuoka Institute of Health and Environmental Sciences, Fukuoka 8180135, Japan; 4Department of Clinical Laboratory, Clinical Research Institute, National Hospital Organization Kyushu Medical Center, Fukuoka 8108563, Japan

**Keywords:** SARS-CoV-2, immunocompromised host, Bruton’s tyrosine kinase, persistent infection

## Abstract

Coronavirus disease 2019 (COVID-19) is an emerging infectious disease caused by severe acute respiratory syndrome 2 (SARS-CoV-2). There are many unknowns regarding the handling of long-term SARS-CoV-2 infections in immunocompromised patients. Here, we describe the lethal disease course in a SARS-CoV-2-infected patient during Bruton’s tyrosine kinase inhibitor therapy. We performed whole-genome analysis using samples obtained during the course of the disease in a 63-year-old woman who was diagnosed with intraocular malignant lymphoma of the right eye in 2012. She had received treatment since the diagnosis. An autologous transplant was performed in 2020, but she experienced a worsening of the primary disease 26 days before she was diagnosed with a positive SARS-CoV-2 RT-PCR. Tirabrutinib was administered for the primary disease. A cluster of COVID-19 infections occurred in the hematological ward while the patient was hospitalized, and she became infected on day 0. During the course of the disease, she experienced repeated remission exacerbations of COVID-19 pneumonia and eventually died on day 204. SARS-CoV-2 whole-viral sequencing revealed that the patient shed the virus long-term. Viral infectivity studies confirmed infectious virus on day 189, suggesting that the patient might be still infectious. This case report describes the duration and viral genetic evaluation of a patient with malignant lymphoma who developed SARS-CoV-2 infection during Bruton’s tyrosine kinase inhibitor therapy and in whom the infection persisted for over 6 months.

## 1. Introduction

Despite the introduction of coronavirus disease 2019 (COVID-19) mRNA vaccines, severe acute respiratory syndrome coronavirus 2 (SARS-CoV-2) continues to mutate and remains as major threat to immunocompromised hosts. Immunocompromised people have an increased risk of developing severe COVID-19 outcomes [1] and might not acquire the same level of protection from COVID-19 vaccines compared with immunocompetent hosts [2]. The adverse outcomes associated with COVID-19 infection in patients with hematological malignancies stem at least in part from intrinsic immune dysfunction [3]. Hematological patients experience delayed viral clearance, which results in persistent shedding of viable SARS-CoV-2 and the emergence of multiple mutations [4].

Whether and for how long immunocompromised patients shed infectious virus has profound implications for understanding disease transmission and treatment for hematological patents. A recent case report showed prolonged infectious SARS-CoV-2 shedding for 143 days post-symptom onset in an immunocompromised patient [5]. However, the detection of viral genomic material does not confirm the presence of infectious SARS-CoV-2.

The duration of infectiousness and, consequently, the necessary duration of isolation in healthcare institutions, are unanswered questions. The United States Centers for Diseases Control and Prevention recommends a 5 day isolation period for COVID-19 patients with mild illness who have been fever free for at least 24 h. This period is extended to up to 10 days for patients with severe infection and/or severe immunosuppression. Ending isolation without a viral test may not be an option in these cases [6].

Here, we describe the lethal disease course in a persistently SARS-CoV-2-infected patient for more than 6 months during Bruton’s tyrosine kinase (BTK) inhibitor therapy. We performed whole-genome analysis using samples obtained during the course of the patient’s disease.

## 2. Materials and Methods

### 2.1. Saliva and Nasopharyngeal Reverse Transcription Polymerase Chain Reaction (RT-PCR) Testing

Salivary or nasopharyngeal samples were collected from the patient, as follows: T1 (Day 0), T2 (Day 34), T3 (Day 43), T4 (Day 49), T5 (Day 55), T6 (Day 78), T7 (Day 107), T8 (Day 114), T9 (Day 128), T10 (Day 135), T11 (Day 140), and T12 (Day 189). The samples underwent RT-PCR testing for SARS-CoV-2 with the Ampdirect™ 2019-nCoV detection kit (Shimadzu Corporation, Kyoto, Japan), in which two sequences specific to SARS-CoV-2, N1 and N2, as defined by the United States Centers for Disease Control and Prevention, were targeted as primers and probes [7]. The real-time PCR analyzer for the SARS-CoV-2 diagnosis was the cobas^®^ z 480 (Roche Diagnostics, Basel Switzerland) or the QIAamp Viral RNA Mini QIAcube Kit (Qiqgen GMBH, Hilden, Germany) which showed that N1 and N2 were amplified, with cycle threshold values of approximately 40. All samples were frozen at −80 °C after the SARS-CoV-2 RT-PCR test was performed.

### 2.2. Serological Testing for SARS-CoV-2 Antibody

Serological testing for antibodies targeting the S1 subunit of the viral spike protein (Immunoglobulin G, spike protein receptor-binding domain) and antibodies targeting the viral nucleocapsid protein (Immunoglobulin G, nucleocapsid protein) was performed at Roche using the Elecsys^®^ Anti-SARS-CoV-2 S test and Elecsys^®^ Anti-SARS-CoV-2 test (Roche Diagnostics), respectively.

### 2.3. SARS-CoV-2 Whole Viral Sequencing and Phylogenetic Analysis

The nucleotide sequences of SARS-CoV-2 obtained from nasopharyngeal swabs or saliva collected from the patient were evaluated at nine time points, as follows: T1 (Day 0), T2 (Day 34), T3 (Day 43), T4 (Day 49), T5 (Day 55), T9 (Day 128), T10 (Day 135), T11 (Day 140), and T12 (Day 189). The whole-genome sequencing of SARS-CoV-2 was performed using the method of Itokawa et al. [8] based on the ARTIC Network’s multiplex PCR (https://artic.network/ncov-2019, accessed on 6 April 2021). The resulting DNA libraries were sequenced with the MiSeq system (Illumina, San Diego, CA, USA).

Sequence reads were analyzed by the method of Sekizuka et al. [9] to obtain the whole genome sequence. The patient’s viral sequences can be located with Global Initiative on Sharing All Influenza Data (GISAID) [10], accession numbers EPI_ISL_15610661, EPI_ISL_15610675, EPI_ISL_15610808, EPI_ISL_15610828, EPI_ISL_15610943, EPI_ISL_15610944, EPI_ISL_15610971, EPI_ISL_15610972, and EPI_ISL_2015094.

The comparison dataset comprised 27 representative SARS-CoV-2 genomes selected from Nextstrain (https://nextstrain.org/, accessed on 17 August 2022) by regions of interest (Japan, Asia, Europe, North and South America). Furthermore, 30 sequences representative of Fukuoka Prefecture, Japan, were selected from the GISAID database.

These reference sequences and those obtained from the patient, in addition to Wuhan-Hu-1, NC_045512.2 were aligned using multiple alignment using fast Fourier transform (MAFFT) software ver.7 (https://mafft.cbrc.jp/alignment/software/tips.html, accessed on 23 August 2022). The best-fit substitution model for the maximum likelihood method was selected using MEGA.11 software (https://www.megasoftware.net/, accessed on 23 August 2022). Maximum likelihood phylogenetic trees were created using the general time reversible plus G substitution (GTR+G) model with 1000 bootstrap replicates (Appendix A).

### 2.4. SARS-CoV-2 Viral Infectivity Studies

Specimens were centrifuged at 3000 rpm for 15 min. The supernatant was filtered through a 0.22 μm filter (Millipore; Sigma-Aldrich, Madrid, Spain) then diluted 1:3 in Dulbecco’s modified Eagle’s medium (D-MEM) (Wako, Osaka, Japan). Next, 50 μL of the diluted sample was added to Vero-E6/TMPRESS cells (Japanese Collection of Research Bioresources Cell Bank; https://cellbank.nibiohn.go.jp/english/, accessed on 23 August 2022) in a 12 well plate. The plate was then incubated for 1 h at 37 °C in a 5% carbon dioxide incubator. After 1 h, the inoculum was removed and washed twice with D-MEM, and then D-MEM with 2% of fetal bovine serum was added at 37 °C in a 5% carbon dioxide incubator. Cells were then incubated for an additional 9 days.

## 3. Case

The patient, a 63-year-old female, was diagnosed with intraocular malignant lymphoma of the right eye in 2012 and began treatment. The lesion later spread to the central nervous system (CNS), and she was treated continuously throughout repeated hospital admissions and discharges. An autologous transplant was performed in 2020 to control the disease. She was later admitted to the hematology ward for rescue therapy owing to a worsening of the primary disease 26 days before she was diagnosed with a positive SARS-CoV-2 RT-PCR. Tirabrutinib was administered for the primary disease on the same day, with a partial response. A cluster of COVID-19 infections occurred in the hematological ward while the patient was hospitalized. At this point, her SARS-CoV-2 RT-PCR test result was negative; however, a fever and sore throat developed 4 days later, and the test result was positive on day 0. We define her initial day of presentation with SARS-CoV-2 RT-PCR positive as day 0 of COIVD-19 infection. Because the designated ward for COVID-19 patients in our hospital had reached its full capacity, the patient was transferred to another hospital for treatment from day 1. Chest computed tomography (CT) was performed at the time of transfer, which showed ground glass opacities (GGO) in the upper lobe of the left lung. Tirabrutinib was discontinued from day 3, and the patient was treated with remdesivir for 5 days. She was transferred back to our hospital on day 17 for continued treatment for the malignant lymphoma, as her condition was good despite continued temperatures of 37.2–37.8 °C. Soon after her return, she gradually developed a temperature above 38 °C and hypoxemia. Methylprednisolone (mPSL, 1 mg/kg) was administered on day 28 owing to an immune response after the COVID-19 infection. Chest CT findings on day 30 showed extensive GGO with contractile changes and infiltrative shadows, and the hypoxemia had worsened. On day 32, tocilizumab and mPSL (500 mg) were administered for their anti-inflammatory effects. However, the pneumonia worsened on day 34. Additionally, the SARS-CoV-2 RT-PCR test results had remained positive since the diagnosis. A second course of remdesivir was administered for 5 days from day 34 onwards, considering a relapse of COVID-19. Thereafter, her condition gradually improved, and the steroid dosage was tapered by 10 mg each week. Tirabrutinib was resumed to control the malignant lymphoma. However, on day 70, head magnetic resonance imaging revealed worsening of the CNS lesions of malignant lymphoma, and whole-brain irradiation was initiated for life-saving purposes. The patient’s consciousness gradually improved, and she was able to eat. However, the suspected complication of aspiration pneumonia was discovered; therefore, antimicrobial agents were administered. On day 110, chest CT again showed pneumonia with GGO in both lungs. Interstitial pneumonia due to COVID-19 or drug-induced pneumonia was considered. On the same day, mPSL (250 mg) was re-administered. As the pneumonia improved, prednisolone (1 mg/kg) was maintained thereafter. However, leukoencephalopathy was identified after radiotherapy, and the patient was discharged and returned home on day 147 to receive palliative care. During the patient’s hospitalization, she never tested negative for SARS-CoV-2 with RT-PCR, and she was negative for antibody production (Figure 1). During home care, hypoxemia progressed on day 189. SARS-CoV-2 RT-PCR was performed, and the result was positive. The patient died on day 204 (Figure 1 and Figure 2).

SARS-CoV-2 RT-PCR-positive samples were used to analyze the genetic mutations during the course of the patient’s disease. Twelve samples (T1–12) were analyzed; T6, T7, and T8 samples could not be analyzed owing to low viral load. Whole-genome sequences were examined for mutational transitions compared with the T1 sequences. As a result: T2 sample, three mutations; T3 sample, five mutations; T4 and T5 samples, one mutation; T9, T10, and T11 samples, five mutations; T12 sample, seven mutations (Figure 3). Over time, whole viral sequencing revealed the development of seven mutations. Viral infectivity studies were performed using all 12 samples. Only the T12 sample confirmed the presence of infectious virus.

## 4. Discussion

The patient described in this case report, with intraocular malignant lymphoma and CNS recurrence, contracted COVID-19 while taking tirabrutinib. Thereafter, shedding of viable SARS-CoV-2 persisted for approximately 6 months, with repeated episodes of COVID-19 pneumonia. The result of the SARS-CoV-2 RT-PCR tests performed during the course of the patient’s hospitalization were never negative, and there was no SARS-CoV-2 antibody production. Whole-viral sequencing analysis of the samples during the study suggested that the same virus persisted, with repeated mutations and persistent infectiousness.

There are many reports of immunocompromised persons shedding SARS-CoV-2 for long periods of time [4,5,11,12,13,14]. One of these reported cases is that of Choi et al. [5] who detected SARS-CoV-2 up to 143 days after infection. The reason for the persistent infection was thought to be the repeated administration of immunosuppressive drugs to control severe antiphospholipid antibody syndrome with alveolar hemorrhage. Factors associated with delayed SARS-CoV-2 clearance include, but are not limited to, older age, severe disease, multiple underlying diseases, immunocompromised status, hematological diseases that involve transplantation, and transplantation for solid tumors [15,16]. The following factors were considered to have contributed to the persistent infection in this case: (1) latent immunodeficiency after auto-transplantation, (2) immunosuppressive state induced by oral tirabrutinib, and (3) repeated administration of steroids.

COVID-19 is a biphasic illness with an initial viremic phase and a later effective adaptive immune phase [17]. Therefore, treatment of COVID-19 depends on the phase. It makes sense to administer neutralizing antibody products and antiviral drugs during the viremic phase and anti-inflammatory drugs during the later effective adaptive immune phase. The timing of anti-inflammatory drug administration is especially important because severe COVID-19 is caused by cytokine storms [17]. Some studies describe disadvantages of immunosuppressive therapy and indicate a delay in viral clearance in immunosuppressed conditions [18,19]. Hematological malignancies are biologically heterogeneous with a spectrum of inherent immune impairment that is further exacerbated by disease-directed therapies. Additionally, antibody production capacity is reduced, which makes it difficult to eliminate the virus, resulting in higher mortality rates [3,20].

Our patient was treated with tirabrutinib for refractory primary CNS lymphoma. Ibrutinib was promptly discontinued owing to concerns about B-cell dysfunction when COVID-19 infection was confirmed. Ibrutinib is a potent, covalent inhibitor of BTK, a kinase downstream of the B-cell receptor that is critical for B-cell survival and proliferation [21,22]. BTK inhibitors, which target a wide range of proinflammatory signaling pathways, may play a key role in the management of COVID-19 [23]. However, it has been suggested that these drugs may be unsuitable for viral elimination because they suppress antibody production by inhibiting B-cell function [24]. Another report describes the ability of Burton kinase inhibitor-treated patients to produce antibodies after vaccination against SARS-CoV-2. The response rate for these patients who received active Burton kinase inhibitors was very low (23%) [25]. In our case, the administration of a BTK inhibitor and steroids suppressed the excessive immune response. However, it is possible that the patient was unable to eliminate the virus owing to a failure to produce antibodies against SARS-CoV-2. As a result, although the patient did not develop severe pneumonia, SARS-CoV-2 was not eliminated, and she was considered to be persistently infectious, with the virus continuing to mutate.

Phylogenetic analysis results over 6 months in this case had a common ancestor. The different sequences and phylogeny of the same area detected at the same time suggested persistent infection rather than reinfection. As mentioned above, it was proven that the decreased ability to produce antibodies following treatment with BTK inhibitor and steroids made it difficult for our patient to eliminate SARS-CoV-2, resulting in repeated replication, which led to persistent infection. Furthermore, the fact that cells could be isolated from the specimens prior to death means that infectivity was sustained.

As in this report, uncertainty continues regarding COVID-19 de-isolation of immunocompromised hosts. Patients should be considered infectious and quarantined during the period of viral shedding. Usually, cell culture methods are not used to determine the end of isolation. Viral culture positivity may also not correlate perfectly with transmissibility, although the correlation between culture data and PCR cycle threshold values may help predict infectiousness [26]. Further data are needed to understand the correlation between transmission risk, culture positivity, and PCR cycle threshold values. However, there are various problems with the stability of the test, such as sampling errors. The presence or absence of symptoms is also important, but there is essentially no other way to address the issue of de-isolation than comply with infection control measures. Above all, it is important to vaccinate and administer antibody products to immunocompromised persons as a means of preventing infection [3].

## 5. Conclusions

This case report describes the duration and genetic study of a patient with malignant lymphoma who was infected with SARS-CoV-2 during BTK inhibitor therapy and who sustained persistent infection for over 6 months. This report provides evidence that B-cell function might play a key role in resolving COVID-19 infection.

## Figures and Tables

**Figure 1 medicina-59-00108-f001:**
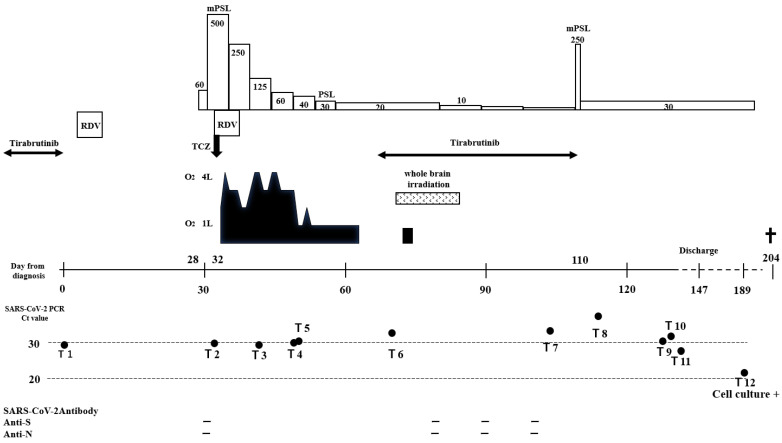
The clinical course of the patient and the time course of the levels of SARS-CoV-2 viral RNA and the results of antibody testing. SARS-CoV-2: severe acute respiratory syndrome coronavirus 2; RDV: remdesivir; TCZ: tocilizumab; mPSL: methylprednisolone; PSL: prednisolone; Ct: cycle threshold, Anti-S; antibody spike protein; Anti-N, antibody nucleocapsid protein.

**Figure 2 medicina-59-00108-f002:**
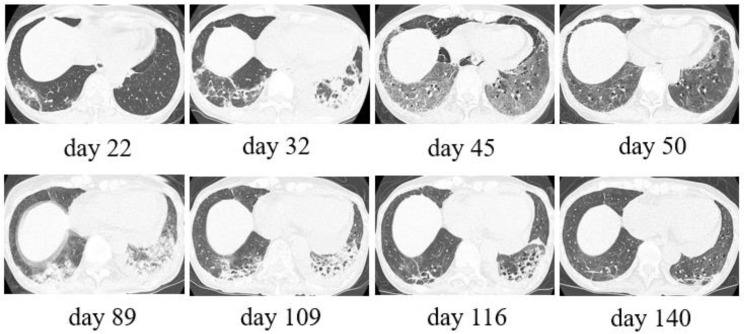
Representative chest computed tomography (CT) image at the level of the lower lobe bronchus.

**Figure 3 medicina-59-00108-f003:**
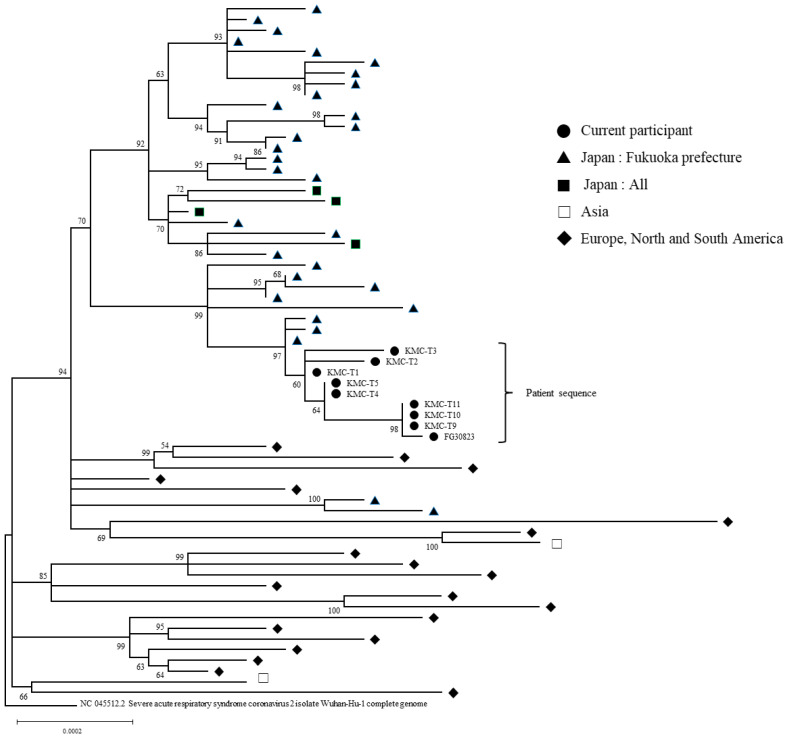
Phylogenetic analysis of SARS-CoV-2 whole-genome virus nucleotide sequences from longitudinally collected nasopharyngeal swab or saliva specimens. Phylogenetic trees were constructed by the maximum likelihood method. Nucleotide sequences of SARS-CoV-2 were obtained from nasopharyngeal swabs or saliva that were collected from the patient at nine time points (●; T1, T2, T3, T4, T5, T9, T10, T11, and T12). Sequences were collected from samples from Fukuoka Prefecture, Japan (▲) and all regions in Japan (■), Asia and Europe (□), and North and South America (◆), as reference sequences. The scale represents 0.0002 nucleotide substitutions per site.

## Data Availability

The data presented in this study are available on request from the corresponding author.

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
