# Peer review of "A Case of a Malignant Lymphoma Patient Persistently Infected with SARS-CoV-2 for More than 6 Months"

_medicina, 2023, doi:10.3390/medicina59010108_

Round 1
Reviewer 1 Report
Dear authors,
The article is interesting, and provide relevant data abou SARS-CoV-2 shedding.
The introduction is well written, as well as the methodology. The methodology strengthens the article which should be puclicated.
I would suggest some minor changes in discussion.
In lines 201, and line 207-210 it is stated that the case report present the longest reported duration, although there are similar evidence in the literature with even longest periods:
please see the following preprint and articles (both available in scientific literature with 218 days and 8 months respectively):
- SARS-CoV-2 shedding, infectivity and evolution in an immunocompromised adult mpatient
Reviewer 2 Report
I thank the academic editor for giving me the opportunity to review this interesting manuscript. This is a case report in which the authors bring to the readers' attention a 63-year-old lady with a previous diagnosis of primary lymphoma of the eye who was undergoing therapy with tyrosine kinase inhibitors.Why was the lady underwent an autologous marrow transplant only in 2020? why was there a 6 year delay from the first diagnosis? I'm having a hard time understanding the numbering of days X+...please be clearer. I don't like the expression "low or high grade fever" because it is confusing: please replace with objective numerical data.
I suggest the authors to implement the discussion with the following papers: Cazzato G, Colagrande A, Cimmino A, Cicco G, Scarcella VS, Tarantino P, Lospalluti L, Romita P, Foti C, Demarco A, Sablone S, Candance PMV, Cicco S, Lettini T, Ingravallo G, Resta L. HMGB1-TIM3-HO1: A New Pathway of Inflammation in Skin of SARS-CoV-2 Patients? A Retrospective Pilot Study. Biomolecules. 2021 Aug 16;11(8):1219. doi: 10.3390/biom11081219. PMID: 34439887; PMCID: PMC8392002.
Gagelmann N, Passamonti F, Wolschke C, Massoud R, Niederwieser C, Adjallé R, Mora B, Ayuk F, Kröger N. Antibody response after vaccination against SARS-CoV-2 in adults with hematological malignancies: a systematic review and meta-analysis. Haematologica. 2022 Aug 1;107(8):1840-1849. doi: 10.3324/haematol.2021.280163. PMID: 34911284; PMCID: PMC9335098.
